# Knowledge, attitudes, practices (KAP) and control of rabies among community households and health practitioners at the human-wildlife interface in Limpopo National Park, Massingir District, Mozambique

**Milton Mapatse**[1]*, **Claude Sabeta**[2¤], **José Fafetine**[1,3], **Darrell Abernethy**[4,5]

**1** Veterinary Faculty, University Eduardo Mondlane, Maputo, Mozambique, **2** Agricultural Research Council-Onderstepoort Veterinary Institute, OIE Rabies Reference Laboratory, Pretoria, South Africa, **3** Centre of Biotechnology (CB-UEM), Maputo, Mozambique, **4** Centre for Veterinary Wildlife Studies, University of Pretoria, Pretoria, South Africa, **5** Aberystwyth School of Veterinary Science, Institute of Biological, Environmental and Rural Sciences, Aberystwyth University, Aberystwyth, United Kingdom

¤ Current address: University of Pretoria, Department of Veterinary Tropical Diseases, Pretoria, South Africa
* miltonkitovet@gmail.com

## Abstract

### Background

Rabies is a viral zoonotic disease that kills more than 26,000 people each year in Africa. In Mozambique, poverty and inadequate surveillance result in gross underreporting and ineffective control of the disease in animals and people. Little is known of the role of human attitudes and behaviour in prevention or control of rabies, thus this study was undertaken to assess the knowledge, attitudes and practices amongst selected households and health practitioners in one affected area, the Limpopo National Park (LNP), Massingir district.

### Methodology

A cross-sectional study was conducted among 233 households in eight villages in LNP and among 42 health practitioners from eight health facilities in Massingir district between 2016 and 2018. Consenting household representatives aged 18 years or over were purposively selected. A KAP survey was administered to obtain information on dog ownership and knowledge of rabies, host species affected, modes of transmission, symptoms, recommended treatment and preventive methods.

Similar to household study participants, health practitioners were purposively selected and completed the questionnaire during the investigators' visit. The questionnaire sought information on knowledge of rabies, management of bite wounds, vaccination sites and schedules of pre- and post-exposure prophylaxis. Descriptive and inferential data analyses were performed using SPSS software version 18.0.

### Results

Approximately twenty per cent (18.9%; 95% CI: 14.3–24.3) and 13.3% (95% CI: 9.4–18.1) of households had good knowledge and practices of rabies, respectively. For health

**Data Availability Statement:** All relevant data are within the manuscript and its Supporting Information files.

**Funding:** This work was supported by the Third Framework Agreement phase III "FA 3 III DGD/ITM (Belgian Directorate General for Development C/ Institute of Tropical Medicine) 2014-2016": Project Communities on the move: animal and human health challenges, awarded to the Faculty of Veterinary Science, University of Pretoria. The funders had no role in study design, data collection and analysis, decision to publish, or preparation of the manuscript.

**Competing interests:** The authors have declared that no competing interests exist.

practitioners, only 16.7% (95% CI: 7.5–31.9) had good knowledge, whilst 33.3% (95% CI: 20.0–49.7) adopted adequate attitudes/practices towards the disease.

## Conclusions/Significance

In conclusion, both households and health practitioners displayed poor levels of knowledge and adopted bad attitudes and practices towards rabies. The former, had more gaps in their attitudes and practices towards the disease. Village location and education level ($P < .05$) and similarly, sex and occupation, were found to be statistically associated with good knowledge of rabies among households as compared to HPs. Overall, a lack of community-based education and professional retraining courses contribute significantly to poor awareness of rabies in the LNP of Mozambique. Enhancing public health knowledge should consequently reduce dog-mediated human rabies deaths in this country.

## Author summary

In Mozambique, rabies is maintained primarily by the domestic dog, the principal vector and host species responsible for the majority of human cases dating back to the early 1900s. Control of animal rabies has historically been undertaken by government veterinary authorities, with limited involvement of the health and environment sectors. In the Massingir District in general, and in Limpopo National Park (LNP) in particular, parenteral dog rabies mass-vaccination campaigns and the provision of post-exposure prophylaxis are inconsistent. Limited resources for dog vaccination campaigns, insufficient veterinary field staff, inefficient disease notification procedures and inadequate training of health practitioners constraint effective rabies control across the country. Awareness of good practices regarding management of bite wounds among local community members and health practitioners is crucial to reducing rabies deaths. The results obtained in this study will inform government policy on practical interventions in the control of dog and human rabies.

## Introduction

Rabies is one of the oldest infectious diseases in medical history, caused by a member of the *Lyssavirus* genus and which can potentially infect all warm-blooded vertebrates, including humans. Today, rabies remains a major public and veterinary health problem in most developing countries, particularly in the tropical and subtropical regions of Africa and Asia [1,2] where it is considered a neglected disease [3,4]. Upon entry into a susceptible host, the highly neurotropic virus induces an acute, progressive and fatal encephalomyelitis, mainly through bites but also, to a lesser extent, through scratches or contact with mucous membranes [5]. The causative agent, responsible for more than 95% of human rabies deaths, is a classical rabies virus (RABV) and one of the 17 members in the genus *Lyssavirus*, family *Rhabdoviridae* [6–8]. Members of this genus are characterized by non-segmented negative-sense RNA genomes of approximately 12 kb in size [9].

Globally, human mortality is estimated to be at least 59,000 deaths per year, 56% of these occurring in Asia and 44% in Africa, especially among rural communal areas with large stray dog populations [10–12] and inconsistent dog mass vaccinations. Poverty and poor awareness

of rabies are generally associated with an increased vulnerability to the disease and are, consequently, major obstacles in prevention and control, especially in rural areas [13,14]. A general understanding about dog behaviour, responsible pet ownership, appropriate health service-seeking behaviour following dog bites and rabies prevention are all crucial in raising public awareness and reducing the number of deaths [15,16].

The National Strategy for the Control of Rabies in Mozambique was approved by the Council of Ministers in 2010 to control rabies in animal populations by limiting its expansion and subsequent spill over into humans. Despite this initiative, rabies deaths continue to be reported annually from all provinces of this country [17,18].

Between 1989 and 2007, 1,001 human cases were reported in Mozambique with a two-fold increase in deaths from 2010 (n = 44) to 2016 (n = 94) and 2017 (n = 89) [19]. Furthermore, in the last four decades, there has been a general population movement to occupy some of the conservation areas, including Massingir [20]. These remote communities now reside at the human-wildlife interfaces with their dogs, exposed to the threat of rabies from wildlife species such as African wild dogs, bat-eared foxes, jackals, mongooses and hyenas [17,21–23]. The risk of rabies was described by Osofsky et al. [24] using the LNP as an example, where the lack of general disease control including the parenteral vaccination of dogs, was of critical importance.

There is a paucity of data in Mozambique on the knowledge of, and attitudes to, rabies [25,26]. Similar to other African countries, children under 15 years are the main victims of rabies, but are not aware of preventive actions following bite contacts with rabies-suspect dogs [27,28]. The majority of victims seek help from traditional healers rather than health treatment centres [29,30] and suspect rabid dogs are destroyed without quarantine [31–33] or no relevant samples are submitted for laboratory confirmation. In Tanzania for example, bite management centres have been set up and have significantly improved the surveillance of rabies in dogs [34,35]. Enhancing public health awareness in the complex socio-ecological systems of protected wildlife conservancies is a prerequisite to addressing the identified gaps, cultural beliefs and behavioural patterns. Similarly, designing relevant public health educational campaigns, supported by appropriate planning, implementation and evaluation of national control programmes [36,37] will assist in the elimination of dog-mediated human rabies. The Mozambique National Health Plan was designed to strengthen primary health care but the lack of adequate health facilities, poor infrastructure and inadequate training of medical staff, particularly on the management of dog bites and rabies cases [38], undermines its objectives. In the Massingir District, the ratio of medical doctors is similar to the rest of the country (1 physician per 10,000 population versus 0.7 per 10,000 respectively; 2018 data), implying that healthcare services are mostly provided by technicians with elementary to intermediate skills in medical care [39,40].

It is crucial that caregivers, both household members and health professionals, have the correct level of knowledge and ability to prevent rabies and manage animal bites when they occur. Hence, the objectives of this study were to determine the awareness, knowledge, attitudes and practices and awareness of rabies, including prevention and control measures applied to the disease on the ground among selected households and healthcare practitioners in a human-wildlife interface area, the Limpopo National Park (LNP).

## Materials and Methods

### Ethics statement

The study was approved by the Biotechnology Centre scientific board (University Eduardo Mondlane, Mozambique) (21.10.16) and by the Animal Ethics Committee of University of Pretoria (V133-16).

## Characteristics of the study area and population

The study was undertaken in the Limpopo National Park, Mozambique, between 2016 and 2018 (Fig 1). The LNP is located to the west of Gaza Province and is delimited by 190 km of fenced international border with South Africa (Kruger National Park) to the west and by both Limpopo (about 260 km) and Elefantes (about 85 km) rivers in the east and south, respectively [41,42]. The LNP includes 13 villages situated along three major rivers, namely the Limpopo River (Macaringue, Munhamane, Cunze, Chipanzo and Maconguele villages); the Shingwedzi River (Bingo, Machamba, Chimangue and Mavoze villages) and Elefantes River (Madingane, Mahlaúle, Chibotane and Macuachane villages). Three villages (Machamba, Bingo and Mavoze) are located inside the national park with the remainder in the Buffer Zone, an area

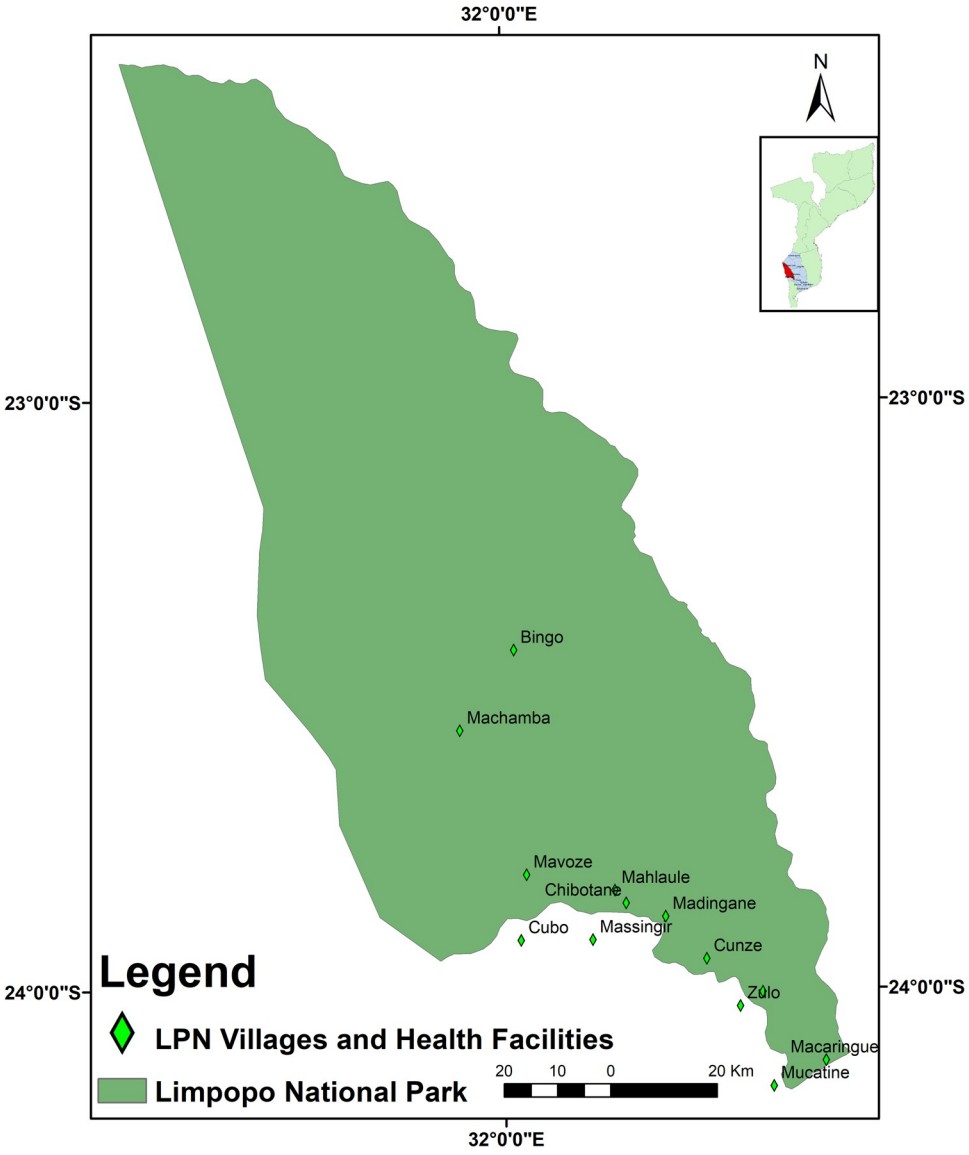

**Fig 1. Massingir District map showing the study area (map provided with permission by Dr Ëlio Muatareque).**

considered to be within the park boundaries. These settlements comprise agricultural lands (crop) and 70% of the livestock [43]. The Buffer Zone extends westwards from the Limpopo River for eight to ten kilometres and northwards from the Elefantes River in the area between the latter's confluence with the Limpopo and Massingir Dam [44].

The human population of the LNP was estimated in 2001 at approximately 27,000 [45] with most inhabitants concentrated in areas with arable alluvial soils [44]. Based on a 2015 census of animals by the District Services of Economic Activities of Massingir, there were approximately 38,000 cattle, 19,000 goats; 7,000 sheep; 2,000 pigs; 1,500 dogs and 51 donkeys in the LNP.

There are eight health facilities in Massingir District, three inside the park, three in the Buffer Zone and two outside.

### Study design and sampling framework

**Household (HHs) survey.** A cross-sectional study was undertaken among 233 households selected within eight of the 13 LNP villages between 2016 and 2018. The villages were selected using stratified multi-stage random sampling. A sampling frame of all the villages was constructed and the "RAND" function in Microsoft Excel 2010 used to generate random entries. The selected villages were Machamba, Bingo and Mavoze inside the park and Macaringue, Mahlaúle, Madingane, Cunze and Munhamane in the Buffer Zone.

**Sample size.** The sample size was calculated using the single population proportion formula as described by Dohoo et al. [46], assuming an estimated weighted measure (87%) of respondents with knowledge of rabies based on two previous community-based cross-sectional studies of rabies awareness in central and northern Mozambique [25,47], a 95% confidence interval, 5% margin of error and 80% power. With 1,968 households in the study area [48], the sample size representative of the Limpopo National Park region was estimated to be 160. The non-response rate was estimated to be 9% and an additional 59 households were added due to positive responses to the interviewer in the villages, providing a total of 233. The weighted sample per village was estimated using the following formula:

w = n*nh/tnh

Where: w = weighted sample per village; n = required sample size (n = 174); nh -total of households per village; tnh–total number of households

**Recruitment and interviews.** Purposive sampling was used to select households. Respondents had to be 18 years of age or older and available to participate at the time of the visit. Face-to-face interviews were conducted in Changana, but were recorded in Portuguese.

**Health practitioner (HPs) survey.** A self-administered survey was conducted among all eight health facilities in Massingir District, Gaza Province (2016 to 2018) and included all 42 working personnel, composed of two doctors, eight physician's assistants, 19 nurses, 11 caregivers and two technicians. Participants were purposively selected [49] among those who deal directly with patients on a daily basis (clinical, nursing and laboratory personnel).

**Data entry and analysis.** Two databases were created for the two study populations in Epi-Info, version 7 (CDC, USA). Data were analysed using SPSS Statistics for Windows software, version 18.0 (SPSS Inc., Chicago, Ill., USA). The analysis focused on descriptive statistics to determine the frequency, proportions, medians and means of the responses regarding both target groups (households and health practitioners).

To assess the dimension of responses (knowledge, attitudes and practices), all potential factors were grouped and scored as in the scoring systems used in similar previous studies [37,50–52], with some slight modifications. The scoring systems used measured the completeness and accuracy of respondents' answers according to the nature of the question.

Depending on the type of question, scores ranged from 0 to 3. For example, for questions on households' knowledge assessment, the maximum overall score stipulated was 13, while it was 10 and 8 respectively, for attitudes and practices. For households, dichotomised variables were categorized as follows: Knowledge into "Poor knowledge" (0–6) and "Good knowledge" (7–13); attitudes into "Negative attitudes" (0–5) and "Positive attitudes" (6–10) and practice scores into "Poor practices" (0–3) and "Good practices" (4–8).

For health practitioners, the maximum overall score was 27 for "knowledge" and five for "attitudes/practices". Knowledge scores regarding health practitioners were categorized into "Poor knowledge" (0–13) and "Good knowledge" (14–27), and attitudes/practices into "Inadequate attitudes/practices" (0–2) and "Adequate attitudes/practices" (3–5).

The overall KAP on rabies (dependent variables) were compared against HH respondents' village, gender, educational level, and occupation, and health post, years of service and occupation (independent variables) for HPs. Inferential analysis was done using the Chi-square test. All analyses were performed using 95% confidence intervals and $P$ value $< .05$ considered as statistically significant.

## Results

### Households characteristics and dog ownership

The socio-demographic characteristics of households are shown in Table 1. The majority of interviewees were male (63.1%). Most participants were aged between 50 and 59 years, comprising 40.3% of study population. Over 60% of respondents had at least a primary school education and 82.8% were farmers. Nine in ten of the households across all villages owned at least

**Table 1. Socio-demographic characteristics of study participants in LNP, Mozambique.**

| Socio demographic characteristics of study participants | Frequency (%) |
|---|---|
| Gender | |
| Male | 147 (63.1) |
| Female | 86 (36.9) |
| Age (years) | |
| $\leq 19$ | 27 (11.6) |
| 20–29 | 34 (14.6) |
| 30–39 | 41 (17.6) |
| 40–49 | 33 (14.2) |
| 50–59 | 94 (40.3) |
| $\geq 60$ | 4 (1.7) |
| Education | |
| None | 85 (36.5) |
| Primary | 141 (60.5) |
| Secondary | 7 (3) |
| Occupation | |
| Farmer | 193 (82.8) |
| Non-farmer | 40 (7.2) |
| Reasons for keeping a dog | |
| Protect crop fields against monkeys | 143 (68.8) |
| Security | 45 (21.6) |
| Herding | 30 (14.4) |
| Hunting | 10 (4.8) |
| As pet | 2 (1) |

one dog (89.3%) and of these, 20.2% of the dogs were sterilized. The most frequent reason for keeping dogs was to provide protection of the crop fields from monkeys (68.8%) while the least popular reason was for a companion pet (1%).

## Knowledge of rabies (Table 2)

The majority of the respondents (97.9%) had heard of rabies and 51.8% of these through family elders. Only twenty-three (10.1%) respondents correctly identified a virus as the causative agent of rabies ($P$ = .0002). The majority of participant households (96.5%) were aware that dogs are the main source of rabies in Mozambique and that dogs could be affected by the disease (98.2%, $P$ = 1.000). There were significantly less respondents with good knowledge who were aware of the zoonotic nature of rabies and the mode of rabies transmissions to humans ($P$ < .05).

Nearly ninety percent (89.9%) of respondents stated they had observed rabies in animals while 1.5% indicated they had seen cases of human rabies. All participants correctly identified rabid dogs as the most common species observed with clinical signs of the disease (Table 2).

Aggression and aimless wandering, accounted for 50% of all observed symptoms and were the most commonly reported signs. Of all study respondents, 82% were aware that rabies is inevitably fatal following the onset of clinical signs ($P$ = .101) (Table 2).

Results on the prevention of rabies are also presented in (Table 2). In brief, 54.8% of respondents mentioned rabies was preventable through parenteral dog vaccination (83.9%, $P$ = .403). Half of the participants were aware of the incurable nature of the disease following onset of symptoms while 25.9% incorrectly thought it was curable through unspecified oral or injectable drugs.

## Attitudes towards rabies health practices (Table 3)

A high proportion (93%) of the respondents said they would euthanize their dog if it were rabid. Significantly more households with positive attitudes would take the same decision. Nearly all (99.1%) were willing to participate in future rabies vaccination campaigns and of these, 95.3% would seek help from health or veterinary authorities if bitten by a suspect rabid dog.

When asked about attitudes adopted for animals with behaviour suggestive of rabies, 96.5% of respondents said that they would either kill or expel them from their homes or neighbourhood, while 3.5% would not take any action. Almost all (96.2%) of the respondents who owned dogs expressed their willingness to register their dogs. Respondents with positive attitudes were more likely to register their dogs compared to those with negative attitudes ($P$ = .106).

## Dog confinement and healthcare practices in case dog bite in households (Table 3)

Of households owning dogs, 80.3% never confined their dogs, while 3.4% restricted them to a yard or enclosure. Households adopting good practices were more likely to confine dogs compared to those adopting poor practices ($P$ = .002). There was no statistical difference in the proportion of households with a single dog (22.2%) that confined them, compared to those with multiple dogs (16.5%) ($P$ = .605).

Most respondents (91%) would first seek help at health centres for the treatment of bite wounds while 4.7% would consult traditional healers. Those adopting poor practices were more likely to seek treatment from traditional healers than those adopting positive practices ($P$ < .0007). At the household level, 64.8% explained that they would wash the bite wounds with water, while 24.5% correctly indicated that they would wash the bite wounds with soapy water.

**Table 2. Knowledge, attitudes and practices among households.**

| Knowledge, attitudes and practices among households | | | | |
|---|---|---|---|---|
| Knowledge | Total | Good | Poor | *P* value |
| | n (%) | n (%) | n (%) | |
| Causative agent of rabies | | | | |
| Virus | 23 (10.1) | 17 (73.9) | 6 (26.1) | .0002 |
| Other/Worm/ psychological problem | 81 (35.5) | 15 (18.5) | 66 (81.5) | |
| Don't known | 124 (54.4) | 12 (9.7) | 112 (90.3) | |
| Animal source of rabies in Mozambique | | | | |
| Dog | 220 (96.5) | 44 (20) | 176 (80) | .359 |
| Livestock (Cattle/Sheep/Goat) | 32 (14) | 18 (56.3) | 14 (43.8) | |
| Cat | 11 (4.8) | 7 (63.6) | 4 (36.4) | |
| All animal species | 4 (1.8) | 0 (0) | 4 (100) | |
| Don't know | 2 (0.9) | 0 (0) | 2 (100) | |
| Affected animals | | | | |
| Dogs | 224 (98.2) | 44 (19.6) | 180 (80.4) | 1.000 |
| Livestock (Cattle, Sheep/Goat) | 31 (13.6) | 15 (48.4) | 16 (51.6) | |
| Cats | 18 (7.9) | 10 (55.6) | 8 (44.4) | |
| Wild animals | 4 (1.8) | 1 (25) | 3 (75) | |
| All animal species | 2 (0.9) | 0 (0) | 2 (100) | |
| Knows about the transmission of rabies to humans | | | | |
| Yes | 189 (82.9) | 43 (22.8) | 146 (77.2) | .008 |
| No | 22 (9.6) | 1 (4.5) | 21 (95.5) | |
| Uncertain | 17 (7.5) | 0 (0) | 17 (100) | |
| Mode of transmission to humans | | | | |
| Bite from infected animal | 167 (88.4) | 42 (25.1) | 125 (74.9) | .03 |
| Contact with infected saliva | 5 (2.6) | 1 (20) | 4 (80) | |
| Don't know | 21 (11.1) | 0 (0) | 21 (100) | |
| Other (Playing with dogs) | 1 (0.4) | 0 (0) | 1 (100) | |
| Ever seen rabid animal | | | | |
| Yes | 205 (89.9) | 44 (21.5) | 161 (78.5) | .01 |
| No | 23 (10.1) | 0 (0) | 23 (100) | |
| Animals seen with rabies | | | | |
| Dog | 205 (100) | 44 (21.5) | 161 (78.5) | |
| Cat | 5 (2.4) | 4 (80) | 1 (20) | |
| Livestock (Cattle/Sheep/Goat) | 6 (2.9) | 3 (50) | 3 (50) | |
| Clinical signs | | | | |
| Aggressiveness | 103 (50.2) | 34 (33) | 69 (67) | .0006 |
| Aimless wandering | 102 (49.8) | 9 (8.8) | 93 (91.2) | |
| Salivation | 41 (20) | 20 (48.8) | 21 (51.2) | |
| Other | 57 (27.8) | 23 (40.4) | 34 (59.6) | |
| Don't know | 3 (1.5) | 1 (33.3) | 2 (66.7) | |
| Recognition about the fatal nature of rabies | | | | |
| Yes | 187 (82) | 41 (21.9) | 146 (78.1) | .101 |
| No | 13 (5.7) | 1 (7.7) | 12 (92.3) | |
| Uncertain | 28 (12.3) | 2 (7.1) | 26 (92.9) | |
| Knows if rabies could be prevented | | | | |
| Yes | 125 (54.8) | 40 (32) | 85 (68) | |
| No | 45 (19.7) | 0 (0) | 45 (100) | |

*(Continued)*

**Table 2.** (Continued)

| Knowledge, attitudes and practices among households | | | | |
|---|---|---|---|---|
| Knowledge | Total | Good | Poor | P value |
| | n (%) | n (%) | n (%) | |
| Unsure | 58 (25.4) | 4 (6.9) | 54 (93.1) | |
| Methods of rabies prevention | | | | |
| Dog vaccination | 104 (83.9) | 37 (35.6) | 67 (64.4) | .403 |
| Killing all suspicious dogs | 3 (2.4) | 1 (33.3) | 2 (66.7) | |
| Confining stray dogs | 1 (0.8) | 0 (0) | 1 (100) | |
| Other (traditional/ears cropping) | 11 (8.9) | 1 (9.1) | 10 (90.9) | |
| Don't know | 5 (4) | 1 (20) | 4 (80) | |
| Knows if rabies is incurable after onset of clinical signs | | | | |
| Yes | 59 (25.9) | 10 (16.9) | 49 (83.1) | .002 |
| No | 114 (50) | 31 (27.2) | 83 (72.8) | |
| Unsure | 55 (24.1) | 3 (5.5) | 52 (94.5) | |

The results on bite cases experienced in the preceding 12 months in households, the respective measures taken by victims are provided in S1 Table.

Of all respondents, 17.6% recalled a family member having been bitten by a dog in the previous 12 months ($P = .208$). A few victims (12.2%) sought post-exposure vaccination. In most cases (56.1%), a nurse provided such treatment, while a traditional healer assisted in four cases (9.8%). Where the respondent knew the status of the dog after the incident (90.2%), in 35.1% of cases the dog had died while in the majority of cases (64.9%), the dogs had remained healthy.

## Animal healthcare practices including dog vaccination in households

The results regarding healthcare and vaccination practices are illustrated as supporting information in S2 Table. Approximately one third (30.8%) of households who owned dogs visited an animal care health centre at least once a year. There was no statistical difference in the proportion of households who visited the animal health officer based on whether they had one or multiple dogs (29.7% versus 31.3% respectively; $P = .829$). The most common reason for visiting an animal health care centre was for dog vaccination (82.8%), while visits prompted by a rabies-suspect dog or for aggressive behaviour accounted for only 9.4%. The extent of dog vaccination (using certificates or rabies antibody titre results as proof of vaccination) differed significantly among villages, with Mavoze having the highest proportion of vaccinated dogs (6%; $P = .001$) and Machamba (0%) the lowest [(data presented as supporting information (S2 Table)]. The most frequent reason for non-vaccination was lack of information of a campaign (62.1%), followed by absence during a campaign (10.1%) and not being aware of the need to vaccinate (9.5%; $P = 1.000$) (S2 Table).

## Level of knowledge, attitudes and practices towards rabies within households

In general, respondents had a good knowledge (18.9%) and practices (13.3%) on rabies, and a positive attitude (87.1%) towards the disease.

Table 4 shows statistically significant differences ($P < .05$) between knowledge of rabies, and location of villages and educational level of the respondents.

**Table 3. Attitudes, dog confinement and healthcare practices in households.**

| Attitudes and practices among households | | | | |
|---|---|---|---|---|
| **Attitudes** | **Total** | **Positive** | **Negative** | ***P* value** |
| | **n (%)** | **n (%)** | **n (%)** | |
| Would allow euthanizing dog if rabid | | | | |
| Yes | 212 (93) | 196 (92.5) | 16 (7.5) | .0001 |
| No | 2 (100) | 0 (0) | 2 (100) | |
| Uncertain | 14 (6.1) | 7 (50) | 7 (50) | |
| Willing to participate in dog vaccination campaigns | | | | |
| Yes | 231 (99.1) | 202 (87.4) | 29 (12.6) | .241 |
| Unsure | 2 (0.9) | 1 (50) | 1 (50) | |
| Would notify to the authorities if bitten by a dog | | | | |
| Yes | 222 (95.3) | 199 (89.6) | 23 (10.4) | .0001 |
| No | 8 (3.4) | 4 (50) | 4 (50) | |
| Unsure | 3 (1.3) | 0 (0) | 3 (10) | |
| Attitudes towards an animal suspected of rabies | | | | |
| Kill/expel it from home or neighbourhood | 220 (96.5) | 195 (88.6) | 25 (11.4) | 1.000 |
| Nothing | 8 (3.5) | 7 (87.5) | 1 (12.5) | |
| Willing to have dogs registered | | | | |
| Yes | 200 (96.2) | 181 (90.5) | 19 (9.5) | .106 |
| No | 2 (1) | 1 (50) | 1 (50) | |
| Unsure | 6 (2.9) | 5 (83.3) | 1 (16.7) | |
| **Practices** | **Total** | **Good** | **Poor** | |
| Dog confinement | | | | |
| Yes | 41 (19.7) | 13 (31.7) | 28 (68.3) | .002 |
| No | 167 (80.3) | 18 (58.1) | 149 (89.2) | |
| Ways of dog confinement | | | | |
| Fenced yard | 5 (2.4) | 0 (0) | 5 (100) | .002 |
| Tied out/chain/runner | 34 (16.3) | 12 (35.3) | 22 (64.7) | |
| Kennel/other type of enclosure | 2 (1) | 1 (0.5) | 1 (0.5) | |
| First place seeking help in case of dog bites | | | | |
| Health centre | 212 (91) | 28 (13.2) | 184 (86.8) | .636 |
| Traditional healer | 11 (4.7) | 2 (18.2) | 9 (81.8) | |
| Home + health centre | 4 (1.7) | 0 (0) | 4 (100) | |
| Home level | 3 (1.3) | 1 (0.4) | 2 (66.7) | |
| None | 3 (1.3) | 0 (0) | 3 (100) | |
| Household level practices for dog bites | | | | |
| Wound wash with water | 151 (64.8) | 9 (6) | 142 (94) | .0007 |
| Wound wash with water + soap | 57 (24.5) | 22 (38.6) | 35 (61.4) | |
| Traditional/spiritual treatment | 13 (5.6) | 0 (0) | 13 (100) | |
| None | 12 (5.2) | 0 (0) | 12 (100) | |
| Take the dog to the veterinarian at least once a year | | | | |
| Yes | 64 (27.5) | 28 (43.8) | 36 (56.3) | .0002 |
| No | 144 (61.8) | 3 (2.1) | 141 (97.9) | |

## Health Practitioners (HPs)

Forty-two medical staff from eight health centres completed the questionnaire. Nurses comprised the greatest proportion (45%) (Fig 2). Twenty-four (57.1%) of the respondents were female. The average length of service was 4.57 years (median: 2 years; range: 1 to 30 years).

**Table 4. Knowledge, attitudes and practices on rabies of household study participants in LNP based on location, gender, age, education and occupation.**

| | Knowledge | | | Attitudes | | | Practices | | |
|---|---|---|---|---|---|---|---|---|---|
| | Good (%) | Poor (%) | *P* value | Positive (%) | Negative (%) | *P* value | Good (%) | Poor (%) | *P* value |
| **Villages** | | | | | | | | | |
| Macaringue | 12 (29.3) | 29 (70.7) | .032 | 35 (85.4) | 6 (14.6) | .910 | 5 (12.2) | 36 (87.8) | .133 |
| Machamba | 6 (26.1) | 17 (73.9) | | 20 (87) | 3 (13) | | 1 (4.3) | 22 (95.7) | |
| Munhamane | 5 (17.2) | 24 (82.8) | | 25 (86.2) | 4 (13.8) | | 5 (17.2) | 24 (82.8) | |
| Mavoze | 12 (27.9) | 31 (72.1) | | 39 (90.7) | 4 (9.3) | | 11 (25.6) | 32 (74.4) | |
| Madingane | 0 (0) | 26 (100) | | 21 (80.8) | 5 (19.2) | | 3 (11.5) | 23 (88.5) | |
| Mahlaúle | 1 (5.9) | 16 (94.1) | | 15 (88.2) | 2 (11.8) | | 1 (5.9) | 16 (94.1) | |
| Cunze | 5 (20) | 20 (80) | | 21 (84) | 4 (16) | | 4 (16) | 21 (84) | |
| Bingo | 3 (10.3) | 26 (89.7) | | 27 (93.1) | 2 (6.9) | | 1 (3.4) | 28 (96.6) | |
| **Gender** | | | | | | | | | |
| Male | 28 (19) | 119 (81) | .934 | 130 (88.4) | 17 (11.6) | .435 | 20 (13.6) | 127 (86.4) | .860 |
| Female | 16 (18.6) | 70 (81.4) | | 73 (84.9) | 13 (15.1) | | 11 (12.8) | 75 (87.2) | |
| **Age in years** | | | | | | | | | |
| $\leq$ 19 | 4 (14.8) | 35 (85.2) | .847 | 24 (88.9) | 3 (11.1) | .931 | 0 (0) | 27 (100) | .142 |
| 20–29 | 6 (17.6) | 28 (82.4) | | 31 (91.2) | 3 (8.8) | | 3 (8.8) | 31 (91.2) | |
| 30–39 | 9 (22) | 32 (78) | | 36 (87.8) | 5 (12.2) | | 5 (12.2) | 36 (87.8) | |
| 40–49 | 4 (12.1) | 29 (87.9) | | 28 (84.8) | 5 (15.2) | | 5 (15.2) | 28 (84.8) | |
| 50–59 | 20 (21.3) | 74 (78.7) | | 81 (86.2) | 13 (13.8) | | 18 (19.1) | 76 (80.9) | |
| $\geq$ 60 | 1 (25) | 3 (75) | | 3 (75) | 1 (25) | | 0 (0) | 4 (100) | |
| **Education** | | | | | | | | | |
| Primary | 35 (24.8) | 106 (75.2) | .007 | 123 (87.2) | 18 (12.8) | .562 | 22 (15.6) | 119 (84.4) | .322 |
| Secondary | 2 (28.6) | 5 (71.4) | | 7 (100) | 0 (0) | | 0 (0) | 7 (100) | |
| None | 7 (8.2) | 78 (91.8) | | 73 (85.9) | 12 (14.1) | | 9 (10.6) | 76 (89.4) | |
| **Occupation** | | | | | | | | | |
| Farmer | 36 (18.7) | 157 (81.3) | .843 | 170 (88.1) | 23 (11.9) | .337 | 27 (14) | 166 (86) | .499 |
| Non-farmer | 8 (20) | 32 (80) | | 33 (82.5) | 7 (17.5) | | 4 (10) | 36 (90) | |

## Health practitioners' knowledge of rabies

HPs' knowledge of the causative agent, principal sources of rabies in Mozambique, mode of transmissions, clinical features and incubation period are outlined in the supporting information (S3 Table). In brief, 76.2% of participants correctly identified rabies virus as the causative agent of the disease, while 85.7% correctly stated the dog as the most important source of rabies in Mozambique. Of those who incorrectly identified the source, five were in their first year and one was in the second year of service. When asked about transmission routes of the disease, 88.1% correctly identified dog bites as the important transmission route. However, only 26.2% correctly identified saliva and 2.4% suggested raw meat as a possible threat. Almost all (95.2%) respondents believed they knew symptoms associated with rabies and most (87.5%) knew the disease was fatal. However, only 40% correctly identified the early, flu-like symptoms of rabies in humans. Similarly, 50% of interviewees correctly indicated hydrophobia as a symptom but only 14.3% identified paralysis. In all these cases, there was no significant difference between those with more advanced training (e.g. nurses) and the assistants ($P > .05$).

## Health practitioners' knowledge of wound categorization

Twenty nine (69%) of the 42 interviewees confirmed they knew about the World Health Organization (WHO) classification of rabies bite exposures and, of these, most (65.5%) correctly

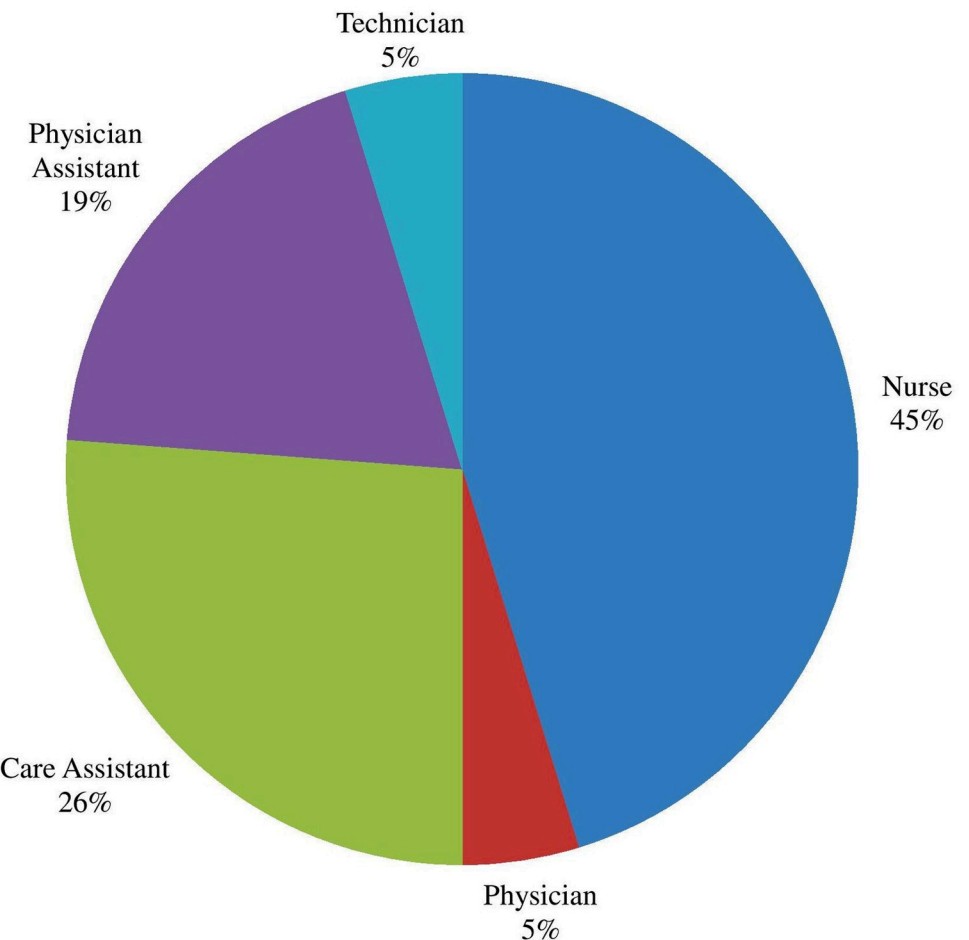

**Fig 2. Professional role of Health practitioner respondents of Massingir district.**

identified mild (Category I) exposure. However, only three (10.3%) correctly identified all the three exposure categories (Table 5). Over half (57.1%) were not aware about the relationship between the site/severity of the bite wound and the period of RABV incubation.

## Health practitioners' practices towards wound bite

Thirty-eight (90.5%) respondents confirmed they knew about wound management. Thirty-seven (97.4%) suggested that bite wounds should be washed with soapy water, while 18.4% felt

**Table 5. Knowledge of wound categorization of health practitioners in Massingir District.**

| | Frequency (%) | | | | Guidelines |
|---|---|---|---|---|---|
| | CAT I | CAT II | CAT III | Don't know | |
| Rabies exposure categories | | | | | |
| Touching or feeding animals, licks on intact skin | 19 (65.5) | 0 | 0 | 23 (35.5) | CAT I |
| Nibbling of uncovered skin, minor scratches/abrasions without bleeding | 0 | 9 (27.6) | 0 | 33 (73.4) | CAT II |
| Single transdermal bites or scratches, licks on broken skin | 0 | 0 | 11 (34.5) | 31 (66.5) | CAT III |
| Multiple transdermal bites or scratches, licks on broken skin | 0 | 0 | 17 (58.6) | 25 (42.4) | |
| Contamination of mucous membranes or broken skin with saliva | 0 | 0 | 8 (27.6) | 34 (73.4) | |

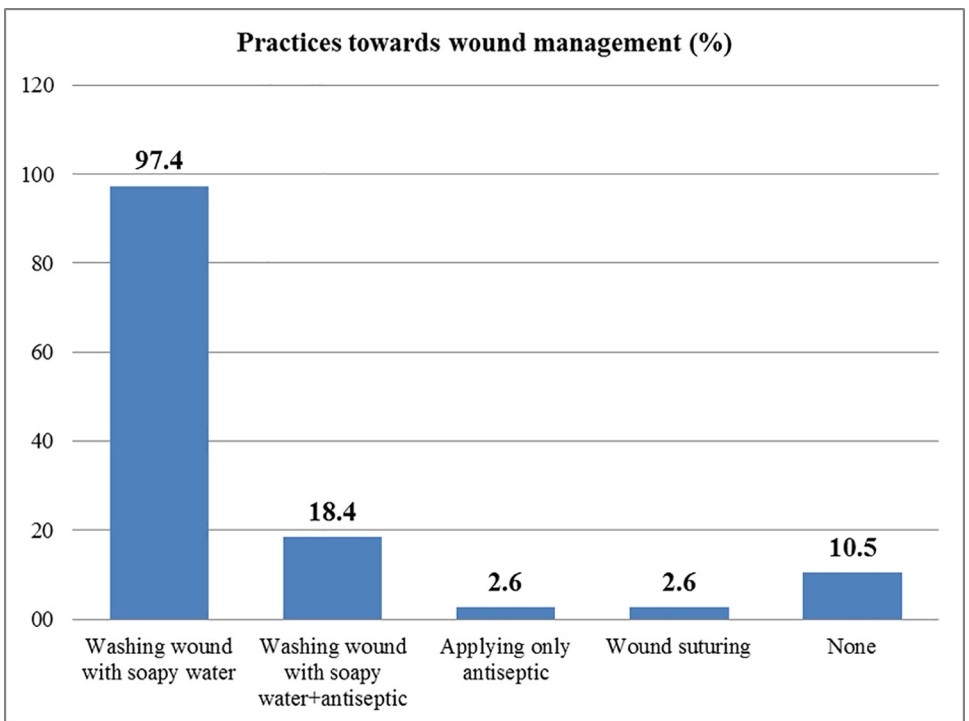

**Fig 3. Health practitioners' practices regarding bite wound.**

that antiseptic was necessary. Interviewees with more than two years' experience (33.3%) were more likely to know this than their less experienced counterparts (12.5%) ($P$ = .136) (Fig 3).

Thirty-four HPs (89.4%) were aware of the advice that should be given to patients exposed to a dog suspected of rabies. However, only thirteen (30.9%) correctly identified isolating the suspect-rabid dogs for a period of up to ten days. A significantly higher proportion (56.3%) of interviewees with more than two years' experience were more likely to know this than their less experienced colleagues (22.2%) ($P$ = .044).

## Knowledge of rabies prevention methods

Regarding rabies prevention, most respondents (83.3%) correctly reported that parenteral dog vaccination was an important preventive measure as dogs were the source of the disease. Fewer study participants identified other useful measures such as destruction of suspect animals (33.3%), public education (26.2%) or pre-exposure prophylaxis (16.7%).

## Knowledge of WHO guidelines for the recommended route(s) and site(s) for the anti-rabies vaccination

Based on the WHO recommendations for the route and sites for administration of the rabies vaccine, 59.5% of HPs stated correctly that the rabies vaccine should be administered via the intramuscular (IM) route, while 21.4% correctly reported intradermal (ID) route as an alternative method. On the other hand, 57.1% of HPs, correctly answered that the site of application of the anti-rabies vaccine was the deltoid muscle and 21.4% also mentioned correctly the thigh as the other region.

**Table 6. Health practitioners who answered correctly about WHO recommended rabies treatment.**

| Variables | CAT I | CAT II | CAT III | Guidelines |
|---|---|---|---|---|
| N | 3 (7.1%) | - | - | CAT I (N) |
| WM+V | - | 7 (16.7%) | - | CAT II (WM+V) |
| WM+V+RIG | - | - | 12 (28.6%) | CAT III (WM+V+RIG) |

N-Nothing; WM-Wound management; V-Vaccine; RIG-Rabies immunoglobulin

### Knowledge of post-exposure prophylaxis (PEP) management according to the category of exposure

When asked about the best strategy to handle Category I, II and III patients, few respondents identified the correct approaches. For Category I– 7.1%, for Category II– 16.7% and for Category III– 28.6% (Table 6).

### Level of knowledge, attitudes/practices within health practitioners

Of the 42 interviewees, 16.7% respondents had a good knowledge, whilst 33.3% adopted adequate attitudes/practices towards the disease.

### Associations between variables under study within health practitioners

The results in Table 7 show statistically significant differences between knowledge of rabies, gender and occupation.

One respondent (100%) of Machamba and Cubo Health posts demonstrated a good knowledge. There were more male respondents who had good knowledge (33.3%) and adopted adequate attitudes/practices (50%) of rabies, than females (4.2 and 20.8%, respectively) ($P$ = .031) (Table 7). The 36–40 year age group was the one with the most respondents (50%) and demonstrated good knowledge of rabies, while for attitudes/practices, best results were observed in the 31–35 year old individuals (66.7%), respectively. With regard to years of service, those with 11–15 years of practice (100%) had good knowledge and, 100% HPs with 11 years onwards adopted adequate attitudes/practices towards rabies. All physicians (100%) and physician assistants (62.5%) were knowledgeable and had adequate attitudes/practices of rabies.

**Table 7. HPs' level of knowledge, attitudes and practices towards rabies, according to location, gender, age, years of service and occupation, in Massingir district, Mozambique.**

| Factors | Knowledge | | | Attitudes/Practices | | |
|---|---|---|---|---|---|---|
| | Good (%) | Poor (%) | *P* value | Adequate (%) | Inadequate (%) | *P* value |
| Villages | | | | | | |
| Macaringue | 0 (0) | 4 (100) | .226 | 2 (50) | 2 (50) | .283 |
| Machamba | 1 (100) | 0 (0) | | 0 (0) | 1 (100) | |
| Massingir-Sede | 5 (22.7) | 17 (77.3) | | 8 (36.4) | 14 (63.6) | |
| Mavoze | 0 (0) | 2 (100) | | 0 (0) | 2 (100) | |
| Cubo | 0 (0) | 1 (100) | | 1 (100) | 0 (0) | |
| Chibotane | 0 (0) | 3 (100) | | 2 (66.7) | 1 (33.3) | |
| Mucatine | 0 (0) | 6 (100) | | 0 (0) | 6 (100) | |
| Zulo | 1 (33.3) | 2 (66.7) | | 1 (33.3) | 2 (66.7) | |
| Gender | | | | | | |
| Male | 6 (33.3) | 12 (66.7) | .031 | 9 (50) | 9 (50) | .096 |
| Female | 1 (4.2) | 23 (95.8) | | 5 (20.8) | 19 (79.2) | |

*(Continued)*

**Table 7.** (Continued)

| Factors | Knowledge | | | Attitudes/Practices | | |
|---|---|---|---|---|---|---|
| | Good (%) | Poor (%) | *P* value | Adequate (%) | Inadequate (%) | *P* value |
| Age group | | | | | | |
| 18–24 | 1 (9.1) | 10 (90.9) | .549 | 2 (18.2) | 9 (81.8) | .396 |
| 25–30 | 4 (16.4) | 20 (83.3) | | 9 (37.5) | 15 (62.5) | |
| 31–35 | 1 (33.3) | 2 (66.7) | | 2 (66.7) | 1 (33.3) | |
| 36–40 | 1 (50) | 1 (50) | | 1 (50) | 1 (50) | |
| >40 | 0 (0) | 2 (100) | | 0 (0) | 2 (100) | |
| Years of service | | | | | | |
| 1–5 | 4 (12.9) | 27(87.1) | .085 | 9 (29) | 22 (71) | .078 |
| 6–10 | 2 (28.6) | 5 (71.4) | | 5 (71.4) | 2 (28.6) | |
| 11–15 | 1 (100) | 0 (0) | | 0 (0) | 1 (100) | |
| > 15 | 0 (0) | 3 (100) | | 0 (0) | 3 (100) | |
| Occupation | | | | | | |
| Physician | 2 (100) | 0 (0) | .019 | 1 (50) | 1 (50) | .241 |
| Physician assistant | 2 (25) | 6 (75) | | 5 (62.5) | 3 (37.5) | |
| Nurse | 2 (10.5) | 17 (89.5) | | 6 (31.6) | 13 (68.4) | |
| Care assistant | 1 (9.1) | 10 (90.9) | | 2 (18.2) | 9 (81.8) | |
| Technician | 0 (0) | 2 (100) | | 0 (0) | 2 (100) | |

## Discussion

Establishing effective rabies prevention and control programs involves an assessment of levels of knowledge, attitudes and practices regarding the disease within local communities [53,54]. As rabies is a preventable disease, it is important that master plans for mitigating the associated risks include education campaigns and assessment of knowledge and adequate practices especially on the mandatory dog vaccination. In Mozambique, for example in 2018, the average vaccination coverage was 8.2% of the three million animals at risk [55], significantly less than the target of 70% set by the WHO to reach adequate herd immunity.

Although the main limitation of this study carried out within the LNP households was the use of non-probabilistic methods for sampling the target population, in general, due to the lack of information and data on the dog population, the scattered location of the houses and households, associated with the constant movement of communities as part of the resettlement underway in the study site, it can nevertheless be considered that the sampled population follows the general demographic patterns of the country. For example, we also found similar demographic profiles to those in a study carried out in Manica (Central Province of Mozambique) [47]. Similar households' demographics characteristics include gender, age, education and occupation. The shortage of health personnel, reflected by the sample size of the present study, follows the established demographic ratio of that of Gaza province, which is 0.07 physicians to 1,000 population [39,40].

The results for the households revealed that there was a low level of rabies awareness within the LNP communities, especially regarding the causative agent, clinical signs and methods of prevention. This worryingly low awareness may be due to the unavailability of general information on the disease from officials and credible sources, including health, veterinary and education authorities. Furthermore, even in the urban areas of Maputo and Matola, educational actions still fall short of what is desirable [32]. If the aim is to meet the target of the endorsed global campaign to end human deaths due to dog-mediated rabies by 2030, awareness and

educational campaigns as a key intervention strategy should be enhanced in the region. World Rabies Day, for example, is an opportune event to disseminate specific messages. The low level of awareness of clinical signs in both animals and humans was also evident. Although death is the final outcome following the onset of rabies symptoms, greater awareness and community education on early detection could significantly reduce the risk of exposure and enhance communities' abilities to make early and correct decisions in rabies-suspect cases. On the other hand, this lack of information may explain why only a minority of respondents knew that they should wash bite wounds with water and soap, confine their dogs and above all, to frequently take their pets to the veterinarian or animal health officer, especially for rabies vaccination. This contrasts with that reported in Ethiopia where the majority of respondents (92.4%) indicated that they would wash their wounds with soap and water [56].

Nevertheless, there was a high level of awareness on some variables related to the animal species that are potential sources of the disease, animals that are susceptible to rabies infection, the zoonotic and fatal nature of rabies, the mode of transmission to humans and prevention methods through vaccination. These high levels of awareness are due to that in recent times, even with the implementation of resettlement programmes, there has been greater access of this population to several means of communication and education (radio, television, mobile phones and educational establishments). The village of Mavoze is an example of this rapid urbanisation. The relationship between high levels of awareness on specific issues previously mentioned and the fact that most respondents had a primary school education cannot be underestimated. In LNP and similar to what was reported in Ethiopia by Alie et al. [57] and Digafe et al. [58], the elders in the family play a key role in disseminating knowledge about rabies. Therefore, education campaigns should seek to better educate elders or provide alternative means of raising awareness. Despite this, and apart from domestic dogs, the role of other species, in particular wild carnivores, was least known. This is a worrying finding given the proximity of these villages to human-wildlife interfaces. Although >75% of respondents were aware of vaccination of dogs as a preventive measure, this did not translate into action given that only 20% of respondents were able to provide proof in the form of a vaccination certificate or other form for the preceding year. This was consistent with other published studies such as in Kenya [59], Sri Lanka [60] and Rwanda [61] where although between 73.9% to 88.1% of the respondents were aware of rabies prevention, their willingness to do so did not guarantee that they could adhere to the different rabies control and prevention campaigns. Regardless of the reasons behind the low proportion of vaccinated dogs, a minimum target of 70% vaccination coverage combined with PEP are necessary prerequisites to eliminate dog-mediated human rabies deaths by 2030. Hence, the importance of joint action of local government and various partners for enhancement of public awareness through community education, for example, via radio, television and community meetings. These public awareness events should underscore the importance of vaccinating dogs as well as the provision of free or subsidised vaccines for use in rural areas where the majority of the population cannot cover vaccination costs.

The fact that most respondents demonstrated positive attitudes about euthanasia of dogs suspected of rabies, including their willingness to participate in future vaccination campaigns and to notify the authorities in cases of bite contacts, in sharp contrast to their normal attitudes/behaviour. These include killing or expelling rabies-suspect dogs from their homes and not checking the vaccination status of their dogs. Furthermore, the long distances they have to travel to the nearest health centre for notification and also for treatment of cases of bite wounds can overshadow the will demonstrated here. Hence, decentralisation and expansion of health and veterinary services should be considered as a priority. Destruction or killing of rabies-suspect animals is not unique to Mozambique but has also been reported in Kenya, Tanzania and the Democratic Republic of Congo [59,62,63]. The negative attitude leading to

such practices greatly contributes to under-reporting of rabies cases as carcasses are not always kept for immediate transport and submission for laboratory testing and confirmation. Cleaveland et al. [62] also pointed out that the elimination of rabies-suspect dogs reduces the number of people who should receive the PEP.

It is therefore necessary to ensure a better allocation of active and passive surveillance teams and logistical conditions for the collection and packaging of samples for rabies testing in these communities. Fortunately, since 2016, a laboratory was set up in the Limpopo National Park, which in addition to accommodating researchers and students, serves as a field station for the preparation of collected samples for further analysis in Maputo. On the other hand, households whose are willing to have their dogs registered may be assisted in this regard in an activity to be carried out simultaneously with the vaccination campaign, provided that the fees are affordable for the LNP population. Such cost-free or even low-fee strategies for dog registration and vaccination have been shown to be an important strategy for rabies control in Bohol, Philippines [64].

In LNP, communities gather, with their livestock and dogs, in their crop fields or for animal health-related events such as vaccination and acaricidal baths points. These points should be considered, together with churches and schools, as strategic places for the implementation of community-based awareness and education campaigns.

Most LNP dogs spend their time roaming freely, foraging for food, but always return to their household of origin. The low socio-economic status, cultural and rural settlement (houses with no fences) characteristics of the LNP communities also mitigate against their confinement. These practices are similar to those reported elsewhere, for example, in Kenya (69%) [65,66], 79% in Madagascar [67] and 55% in Mali [68]. Confinement, rather than a measure of responsible dog ownership, can also minimize the risk of contracting or spreading rabies, in addition to helping to plan more effective parenteral vaccination campaigns. For instance in 2020, forty-one people were reported to have been bitten, mainly by neighbours' dogs. This scenario is illustrative of the risk that non-confinement can bring in spreading the disease. Moreover, the majority only stated that they would use water to wash bite wounds, a measure considered insufficient to prevent the disease. Rabies control programmes should therefore take this aspect into account.

The general willingness shown by most respondents to seek treatment from a health centre following dog bite(s) is commendable and a good starting point for mitigating the risk of contracting rabies. However, these actions must be reinforced by effective practices of wound washing with soapy water, including the minimum washing time. In this survey, as also reported from Ethiopia [58,69], few respondents correctly indicated that washing dog-bite wounds with soapy water is a simple first aid measure that substantially reduces (by five-fold) the risk of contracting rabies [70].

Another objective of our investigation was to raise awareness and subsequently inform politicians and managers in the health sectors about the need for further training and strengthening the skills of HPs on prevention and PrEP/PEP guidelines. The implementation of a One-Health approach to not only rabies, but to other zoonotic diseases, cannot be over-emphasised. The study revealed that nearly a quarter of HPs respondents were not aware of the causative agent of rabies and the major animal source of the disease in Mozambique. Other studies obtained similar results for the "viral" cause (62%-77.5%) in India and in Pakistan [71,72] respectively, while awareness of the dog as the primary vector species responsible for human rabies cases was higher in India (n = 100) [73] compared to this study (98% versus 85.7%). For a health professional, mastery of such concepts is essential as they underpin correct decision-making on optimal therapeutic or prophylactic approaches. Again, this underscores the need for training, retraining courses and capacity building actions. Online courses available on the

Global Alliance for Rabies Control Education Platform (https://rabiesalliance.org) are valuable and could be utilised.

It is also crucial that health professionals be alert to any case reports of patients with a history of wound bites, scratches, or other contact with a potentially infectious vector. Apart from death, a few respondents mentioned other clinical signs indicative of encephalitis due to rabies. Our findings differed from those of Kishore et al. [52], where hydrophobia was one of the clinical signs most recognized by the health workers of Uttarakhand, India (95.7%). Symptoms of rabies in people may be similar to other encephalitic diseases such as malaria and HPs must consider the former as a differential diagnosis. Mallewa et al. [74] in Malawi, pointed out that a central nervous system infection should not only be attributed to cerebral malaria, although most of the clinical signs are similar to those of rabies. Wound management and rabies PEP depend on appropriate pre-assessment and correct wound classification [75], and the lack of general awareness among HPs in this regard may delay or worsen timeous and appropriate post-exposure measures. The low proportion of HPs who could correctly advise the monitoring of biting animals for ten days may be due to the fact that most LNP houses are not fenced. In addition, the apparent lack of communication and co-ordination between HPs and animal health officers is an Achilles heel in the confinement and monitoring of the suspect animals. Veterinary personnel are therefore expected to confine and closely monitor rabies-suspect animals or even those responsible for human bite contact. The lack of a One Health approach in advice given by the HPs to patients was also observed by Malhotra et al. [76], in India, and by Niang et al. [77] in Senegal.

The association between the site of bite and the wound severity (scratch versus deep skin penetration) is likely to affect the outcome of clinical rabies [30]. In the present study, there was apparently a lack of perception of this association. Furthermore, a considerable percentage of study participants was not aware of guidelines on both the site and route of rabies vaccinations, including the pre- and post-exposure prophylaxis scheme. The introduction of refresher and re-training courses on neglected diseases such as rabies may improve their understanding. In Massingir District, less than half of HPs had specific background training in nursing and lacked access to practical training on anti-viral prophylaxis. The weak and slow development of the health sector, including the scarcity of rabies vaccine and rabies immunoglobulins, may also influence the lack of awareness of its importance and use within the health staff. Similarly in India [78] it was evident that even physicians were not adequately informed about the importance of administering vaccines and human rabies immunoglobulins. In 2004, MISAU (Ministry of Health) launched a rabies handbook edited by Barreto et al.[79]. In this handbook, all the basic information about the disease was described for health professionals to understand and apply the knowledge. Surprisingly, the health practitioners still have gaps in their knowledge of rabies and its prevention, as well as in the management of bites.

## Conclusions and recommendations

Based on findings from this study, we conclude that the rural communities of LNP displayed a poor level of knowledge and adopted bad practices towards rabies and this may increase the risk of contracting rabies in the community. Health practitioners had poor knowledge, attitudes and practices on the control and prevention of rabies.

The low level of education and the remote location of the communities were major factors contributing to the poor awareness among households. Medical retraining courses, local education and public awareness on rabies prevention and control, and the management of bite wounds, should be implemented in the short term by LNP veterinary and health authorities especially if the plan is to end human deaths due to dog-mediated rabies by 2030.

The results presented in this study should assist LNP governmental and non-governmental entities in continuously improving rabies prevention and control programmes, including permanent community education and awareness campaigns on the importance of adopting responsible dog ownership and adherence to regular parenteral dog rabies vaccination campaigns. The findings also emphasise the need to provide training and refresher courses for professionals regarding pre- and post-exposure prophylaxis measures and furthermore, to encourage exposed patients to confine rabies-suspect animals and seek both medical and veterinary assistance.

## Study limitations

A major limitation of this study is the tension in LNP over the resettlement process that dates back to 2006. This process created animosity, mistrust and a general lack of openness to any individual who intends to work in such communities (Milgroom and Spierenburg [80]). Furthermore, a low number of health professionals were interviewed, directly reflecting the shortage of human resources in the Massingir district health sector. The shortage of human resources has delayed the efforts of the Government of Mozambique to expand and improve the provision of basic health services.

## Supporting information

**S1 Table. Healthcare practices after dog bite in households in the preceding 12 months.**
(DOCX)

**S2 Table. Frequency of dog vaccination per village and reasons for non-vaccinating against rabies.**
(DOCX)

**S3 Table. Knowledge of rabies causative agent, main animal source of rabies inMozambique, mode of transmissions, clinical features, incubation and concept ofrabies categories of exposure among Health practitioners, Massingir district.**
(DOCX)

## Acknowledgments

We are grateful to Dr Sérgio Salomão Bié (Limpopo National Park) for providing some data related to the Limpopo National Park, to Dr Guilhermina Sitoe (Former Health Director of the Massingir District) for allowing us to work at health post/centres of Massingir District. We are thankful to the communities of all the villages and to the health workers of all post and health centers in Massingir, especially to those who orally consented to participate in this study. We thank the Biotechnology Centre for the logistics and to dr Remígio Mungói (District Services for Economic Activities of Massingir), Dr Vlademiro Magaia, Dr Ofélia Nhambirre, André Nhambir and Iara Gomes for their help in the field. We are grateful to Dr Abel Chilundo, Dr Adilson Bauhofer and Professor Alberto Pondja for assisting in the insertion, organisation and interpretation of the statistical data. A special thanks goes to Dr Élio Muatareque who worked on and allowed us to use the Massingir district map.

## Author Contributions

**Conceptualization:** Milton Mapatse, Claude Sabeta, Darrell Abernethy.

**Data curation:** Darrell Abernethy.

**Formal analysis:** Claude Sabeta, Darrell Abernethy.

**Funding acquisition:** José Fafetine.

**Investigation:** Milton Mapatse.

**Methodology:** Darrell Abernethy.

**Project administration:** José Fafetine.

**Supervision:** Claude Sabeta, José Fafetine, Darrell Abernethy.

**Validation:** Darrell Abernethy.

**Writing – original draft:** Milton Mapatse.

**Writing – review & editing:** Milton Mapatse, Claude Sabeta, José Fafetine, Darrell Abernethy.

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
