## [Decision Letter · Decision Letter 0]

6 Dec 2020

Dear Dr. Mapatse,

Thank you very much for submitting your manuscript "Knowledge, attitudes, practices (KAPs) and control of rabies among community households and health practitioners at the human-wildlife interface in Limpopo National Park, Massingir District, Mozambique" for consideration at PLOS Neglected Tropical Diseases. As with all papers reviewed by the journal, your manuscript was reviewed by members of the editorial board and by several independent reviewers. In light of the reviews (below this email), we would like to invite the resubmission of a significantly-revised version that takes into account the reviewers' comments. 

We cannot make any decision about publication until we have seen the revised manuscript and your response to the reviewers' comments. Your revised manuscript is also likely to be sent to reviewers for further evaluation.

Sincerely,

Daniel Leo Horton, PhD

Associate Editor

David Harley

Deputy Editor

Reviewer's Responses to Questions

**Key Review Criteria Required for Acceptance?**

**Methods**

-Are the objectives of the study clearly articulated with a clear testable hypothesis stated?

-Is the study design appropriate to address the stated objectives?

-Is the population clearly described and appropriate for the hypothesis being tested?

-Is the sample size sufficient to ensure adequate power to address the hypothesis being tested?

-Were correct statistical analysis used to support conclusions?

-Are there concerns about ethical or regulatory requirements being met?

Reviewer #1: The objective should be refined, its too long delete unnecessary words. I haven't seen the testable hypothesis in the manuscript. The study design is appropriate , well planned by the investigator. these two types of population are clearly discussed in the document. The sample size is fine but unfortunately no hypothesis is tested in this study. looking at the abstract, methodology section . what criteria did you use choose the households please include it in the abstract. still under the abstract the data analysis tool is not mentioned . i suggest we should have a subtitle for ethical or regulatory requirements. According line 207,What is that model proposed elsewhere? we need it to know that name of that model.

Reviewer #2: (No Response)

**Results**

-Does the analysis presented match the analysis plan?

-Are the results clearly and completely presented?

-Are the figures (Tables, Images) of sufficient quality for clarity?

Reviewer #1: line 34, statements such as in general are vague. Does such a finding reflect on the entire population. More work needs to be done, some results can be put in a tabular form. and concentrate on the key findings that you think can bring out the practical aspect, that can advocate for policy formulation and control of rabies in LNP. Line 313 & 314 Global results regarding the level of knowledge, attitudes and practices towards rabies within households. is this true? Are you conducting this research on a global scale or in LNP ? Please throw more light on this subtopic.

Reviewer #2: (No Response)

**Conclusions**

-Are the conclusions supported by the data presented?

-Are the limitations of analysis clearly described?

-Do the authors discuss how these data can be helpful to advance our understanding of the topic under study?

-Is public health relevance addressed?

Reviewer #1: The results and conclusions are disconnected. Please ensure that your document flows logically. Discussion and conclusion line 437 & 438 the emphasis is more about livestock. Please consider the public health relevance in your discussion, since your study is about rabies at human-wildlife interphase.

Reviewer #2: (No Response)

**Editorial and Data Presentation Modifications?**

Reviewer #1: (No Response)

Reviewer #2: (No Response)

**Summary and General Comments**

Reviewer #1: This is good work and can be publishable , the abstract should be written well to reflect the major components of the study. clearly outline the research ethics let it stand out. reduce on the wording in the entire document. Major revisions should be done.

Reviewer #2: This is a significant study particularly in Africa where rabies is prevalent. However, I have some issues on the sample size and method on respondent selection. The authors need to describe them fully in the paper. In addition, the paper needs to be fully proofread to be acceptable for publication.

PLOS authors have the option to publish the peer review history of their article (what does this mean?). If published, this will include your full peer review and any attached files.

Reviewer #1: Yes: Omodo Michael

Reviewer #2: Yes: Gino C. Matibag
---

## [Decision Letter · Decision Letter 1]

9 Apr 2021

Dear Dr. Mapatse,

Thank you very much for submitting your manuscript "Knowledge, attitudes, practices (KAPs) and control of rabies among community households and health practitioners at the human-wildlife interface in Limpopo National Park, Massingir District, Mozambique" for consideration at PLOS Neglected Tropical Diseases. As with all papers reviewed by the journal, your manuscript was reviewed by members of the editorial board and by several independent reviewers. In light of the reviews (below this email), we would like to invite the resubmission of a significantly-revised version that takes into account the reviewers' comments. Please ensure you respond to reviewers comments in the attachment

We cannot make any decision about publication until we have seen the revised manuscript and your response to the reviewers' comments. Your revised manuscript is also likely to be sent to reviewers for further evaluation.

Sincerely,

Daniel Leo Horton, PhD

Associate Editor

David Harley

Deputy Editor

Reviewer's Responses to Questions

**Key Review Criteria Required for Acceptance?**

**Methods**

-Are the objectives of the study clearly articulated with a clear testable hypothesis stated?

-Is the study design appropriate to address the stated objectives?

-Is the population clearly described and appropriate for the hypothesis being tested?

-Is the sample size sufficient to ensure adequate power to address the hypothesis being tested?

-Were correct statistical analysis used to support conclusions?

-Are there concerns about ethical or regulatory requirements being met?

Reviewer #1: This study assessed the knowledge, attitudes and practices towards rabies among selected households and health practitioners in Limpopo National Park, Massingir district, Mozambique. Need to improve on this objective terms like control are not captured in the abstract at least reconstruct the objective in the abstract. The number of household + medical setting samples captured in this study are a small size, why such a small sample , clearly explain the type of sampling was it purposive or convenient sampling. And justify for the small sample size(households + medical settings) obtained in this study . Do you think these findings can be used by the authority to implement rabies control programs in River LNP if so explain. Being a national park i expected that may be we have other rabies virus transmitters in the wild that have attacked humans and animals apart from domestic dogs. 

I have looked through the document, the results and conclusions are clearly presented, my concern as scientist before you set out to conduct this study. 1) what was your research question or hypothesis that you want to prove right or wrong about rabies in Limpopo national park. concerns about ethical or regulatory requirements were met.

Reviewer #2: I appreciate the efforts of the authors in improving the paper from the original manuscript.

However, there is still much work to be done here to be acceptable for publication.

1. It is difficult to read. One can hardly grasp the idea of the paper because of so many typo and clerical errors.

2. There is inconsistency in the presentation of numerical data in the narrative.

3. The Tables are not of sufficient quality to be acceptable to an average reader.

4. There are issues in the Discussion section that are emphasized to be part of the study but are not shown in the Results section, e.g. sex of the dogs.

If PLOS wants to accept this paper for publication, another major revision needs to happen.

**Results**

-Does the analysis presented match the analysis plan?

-Are the results clearly and completely presented?

-Are the figures (Tables, Images) of sufficient quality for clarity?

Reviewer #1: results are well presented and the analysis presented matches the analysis plan using the epi info versions and SPSS u Statistics for Windows, version 18.0 (SPSS Inc., Chicago, Ill., USA). The figure and tables tables are of sufficient qualit

Reviewer #2: (No Response)

**Conclusions**

-Are the conclusions supported by the data presented?

-Are the limitations of analysis clearly described?

-Do the authors discuss how these data can be helpful to advance our understanding of the topic under study?

-Is public health relevance addressed?

Reviewer #1: Yes the conclusions are supported by the small set of data presented regarding the smaller number of medical professions that is fine. But i dont think the households in the LNP were limited as well. Clear explanation for this need to be presented. The authors have discussed the topic well and have shown how the data collected can be useful if certain situations are improved in the area.

Reviewer #2: (No Response)

**Editorial and Data Presentation Modifications?**

Reviewer #2: (No Response)

**Summary and General Comments**

Reviewer #1: Generally this work is fine well presented and the authors require to address the suggested comments.

Reviewer #2: (No Response)

PLOS authors have the option to publish the peer review history of their article (what does this mean?). If published, this will include your full peer review and any attached files.

Reviewer #1: Yes: Omodo Michael

Reviewer #2: Yes: Gino Matibag
---

## [Decision Letter · Decision Letter 2]

5 Jul 2021

Dear Dr. Mapatse,

Thank you very much for submitting your manuscript "Knowledge, attitudes, practices (KAPs) and control of rabies among community households and health practitioners at the human-wildlife interface in Limpopo National Park, Massingir District, Mozambique" for consideration at PLOS Neglected Tropical Diseases. As with all papers reviewed by the journal, your manuscript was reviewed by members of the editorial board and by several independent reviewers. In light of the reviews (below this email), we would like to invite the resubmission of a significantly-revised version that takes into account the reviewers' comments. 

Further improvements are recommended by reviewers. I recommend you pay particular attention to the suggestions around presentation of the data , in addition to the specific comments in the attached annotated version of the paper

We cannot make any decision about publication until we have seen the revised manuscript and your response to the reviewers' comments. Your revised manuscript is also likely to be sent to reviewers for further evaluation.

Sincerely,

Daniel Leo Horton, PhD

Associate Editor

David Harley

Deputy Editor

Further improvements are recommended by reviewers. I recommend you pay particular attention to the suggestions around presentation of the data , in addition to the specific comments in the attached annotated version of the paper

Reviewer's Responses to Questions

**Key Review Criteria Required for Acceptance?**

**Methods**

-Are the objectives of the study clearly articulated with a clear testable hypothesis stated?

-Is the study design appropriate to address the stated objectives?

-Is the population clearly described and appropriate for the hypothesis being tested?

-Is the sample size sufficient to ensure adequate power to address the hypothesis being tested?

-Were correct statistical analysis used to support conclusions?

-Are there concerns about ethical or regulatory requirements being met?

Reviewer #2: The objectives are clear, methods appropriate. However, due to the smallness of the sample, there needs to be weighting of the sample, which means tweaking of the numbers. Had the sample been robust or met the minimum, a straight forward calculation of the required sample size would have been easy.

**Results**

-Does the analysis presented match the analysis plan?

-Are the results clearly and completely presented?

-Are the figures (Tables, Images) of sufficient quality for clarity?

Reviewer #2: As this is the third or fourth draft of this paper that I have reviewed, I am disappointed to say that although it is better than the earlier versions, the present quality of the presentation is not good for publication at this stage.

For example, some data are better presented in Tables rather than in Figures. Other data are presented in the narrative but could not be found in Tables of Figures.

The Tables are not uniformly well presented. The level of significance are not shown in some Tables (e.g., Table 4 and 5) where they are necessary.

Additional comments are in the paper.

**Conclusions**

-Are the conclusions supported by the data presented?

-Are the limitations of analysis clearly described?

-Do the authors discuss how these data can be helpful to advance our understanding of the topic under study?

-Is public health relevance addressed?

Reviewer #2: The conclusion partially answered the objectives. It is lacking in explanation why HP level of KAP is lower than that of the community. Limitations are described. Relevance to public health is described, however, the paper is poorly organized and presented that the association with public health measures is weak.

**Editorial and Data Presentation Modifications?**

Reviewer #2: There are still major revisions that need to be addressed particularly in the Discussion section.

**Summary and General Comments**

Reviewer #2: There are still major revisions that need to be addressed particularly in the Discussion section.

PLOS authors have the option to publish the peer review history of their article (what does this mean?). If published, this will include your full peer review and any attached files.

Reviewer #2: Yes: Gino C. Matibag
---

## [Editor Report · Decision Letter 3]

25 Jan 2022

Dear Dr. Mapatse,

We are pleased to inform you that your manuscript 'Knowledge, attitudes, practices (KAP) and control of rabies among community households and health practitioners at the human-wildlife interface in Limpopo National Park, Massingir District, Mozambique' has been provisionally accepted for publication in PLOS Neglected Tropical Diseases.

Best regards,

Daniel Leo Horton, PhD

Associate Editor

David Harley

Deputy Editor

---

## [Editor Report · Acceptance letter]

27 Feb 2022

Dear Dr. Mapatse,

We are delighted to inform you that your manuscript, "Knowledge, attitudes, practices (KAP) and control of rabies among community households and health practitioners at the human-wildlife interface in Limpopo National Park, Massingir District, Mozambique," has been formally accepted for publication in PLOS Neglected Tropical Diseases.

Best regards,

Shaden Kamhawi

co-Editor-in-Chief

Paul Brindley

co-Editor-in-Chief
